# Language Preference for Expression of Sentiment for Nepali-English Bilingual Speakers on Social Media

**Niraj Pahari** and **Kazutaka Shimada**
Kyushu Institute of Technology
680-4 Kawazu, Iizuka, Fukuoka 820-8502, Japan
pahari.niraj828@mail.kyutech.jp and shimada@ai.kyutech.ac.jp

## Abstract

Nepali-English code-switching (CS) has been a growing phenomenon in Nepalese society, especially in social media. The code-switching text can be leveraged to understand the socio-linguistic behaviours of the multilingual speakers. Existing studies have attempted to identify the language preference of the multilingual speakers for expressing different emotions using text in different language pairs. In this work, we aim to study the language preference of multilingual Nepali-English CS speakers while expressing sentiment in social media. We create a novel dataset for sentiment analysis using the public Nepali-English code-switched comments in YouTube. After performing the statistical study on the dataset, we find that the proportion of use of Nepali language is higher in negative comments when compared with positive comments, hence concluding the preference for using native language while expressing negative sentiment. Machine learning and transformer-based models are used as the baseline models for the dataset for sentiment classification. The dataset is released publicly.

## 1 Introduction

In recent years, use of social media and computer mediated communication has increased with millions of users everyday. This increase in social media has consequently increased the use of code switching (CS) or code mixing content. CS can be broadly defined as the linguistic behavior of comprehending the language that is composed of lexical items and grammatical structure from two or more languages with no change of the interlocutor or topic. Throughout this paper, we adopt the stance that the terms 'code switching' and 'code mixing' are used interchangeably to refer to the phenomenon of alternating between two or more languages within a single discourse. Although there may be subtle nuances in usage within certain linguistic contexts, for the purpose of our study, both terms are treated as synonymous and describe the same linguistic behavior.

CS was earlier associated with the spoken language, but due to the informal nature of social media, CS is also found in written form (Bali et al., 2014). The language spoken by multilingual individual is closely connected to emotion (Rajagopalan, 2004). Similarly, emotion is a driving factor for CS behaviour (Ndubuisi-Obi et al., 2019). Linguistics researchers have found that multilingual speakers have a certain language of preference for expressing their emotions (Dewaele, 2010; Rudra et al., 2016). Hence, the task of sentiment analysis and socio-linguistic studies based on the sentiment of multilingual speakers have received a lot of attention in the NLP domain. These studies have shed light on different characteristics of the society. Several studies have analyzed the language preference in multilingual societies and concluded that multilingual speakers indeed prefer their first language (L1) while conveying their emotions (Agarwal et al., 2017; Rudra et al., 2019). On the other hand, most studies in the field of code-switching have only focused on the high-resource language pairs. Up to now, far too little attention has been paid to leveraging the growing amount of Nepali-English CS text in social media and analyzing the language preference for the sentiment emotions for Nepalese multilingual community.

Sentiment analysis is a computational technique used to determine the sentiment or emotional attitude conveyed in a text. Sentiment analysis can help in obtaining insights from the opinion on certain products or subjects of interest from the users and help in planning the business strategies (Balage Filho et al., 2012). The applications and resources for sentiment analysis are mostly created for high-resourced languages in monolingual settings. However, the annotated data for monolingual data cannot handle code-switched scenarios and fails to leverage good results (Al-

Ghamdi et al., 2016). Several researchers have constructed the sentiment analysis dataset for code-switched scenarios (Chakravarthi et al., 2020b; Hegde et al., 2022). However, to the best of our knowledge, there is no existing sentiment analysis dataset for code-switched Nepali-English language even though Nepali-English mixed language has emerged as a dialect in the Nepalese community owing to the increasing use of English elements in Nepali conversation.(Gurung, 2019).

In this study, we collect the public comments in code-switched Nepali-English from Youtube platform and annotate them with sentiment annotations. We hypothesize two different hypotheses to analyze the relation between the language used in the comment with the sentiment of the comment and preferred language for expression of Negative or Positive sentiments. The contributions of this study are as follows:

1. We present the first standard code-switched Nepali-English dataset for sentiment analysis.

2. We perform statistical studies to identify the language preference by Nepali-English multilingual speakers in social media.

3. We provide experimental analysis of machine learning- and deep learning-based models on our code-switched dataset for sentiment analysis.

## 2 Related Work

The preferred language for expression of opinions by multilinguals has been studied by linguists for a long time. Fishman (1970), studies the behavior of English-Spanish bilinguals and report the use of English for professional purposes and Spanish for informal purposes like chatting. Barredo (1997) studies the pragmatic functions of Basque-Spanish code-switching and made several conclusions, one of them being: Basque-Spanish multilingual speakers normally switch to Spanish to convey humor and irony. Dewaele (2004) identifies how multilingual speakers highly use their first language for swearing and taboo words. The authors report that the multilingual speakers, while using code-switching/mixing, tend to use their first language for swearing even when the language is not understood by their interlocutor(s). Hindi and Nepali languages are closely related with each other and belong to the same language fam-

ily. For Hindi-English code-switched data, Agarwal et al. (2017) analyze the English-Hindi code-switching and swearing pattern on social networks and conclude that the multilingual speakers have strong preference for swearing in the dominant language. Rudra et al. (2019) study different aspects of English-Hindi code-switching in Twitter and identify the preference of expressing negative sentiments using Hindi language is twice as much as English. In the context of Nepali-English code-switching, the study by Gurung (2019) presents a detailed socio-linguistic study on CS phenomenon in the conversations between Nepalese people. This study studies the extent, role of media, and reason in mixing of Nepali-English languages. To the best of our knowledge, there is no existing study that is focused on studying the language preference in Nepali-English code-switched scenarios.

Computational linguists have been studying code-switching for a substantial period of time. Several data resources have been created for the support of research on code-switching. Solorio et al. (2014) release code-switched dataset for language identification tasks in four language pairs, Nepali-English being one of them. They extract the sentences from social media platforms like Twitter and Facebook. Similarly, Patwa et al. (2020) release the sentiment dataset for code-switched Hindi-English and Spanish-English language pairs. The datasets constitutes of code-switched tweets with sentiment annotation among three classes: Positive, Neutral, and Negative. A considerable amount of literature has been published utilizing Youtube comments as a source of sentiment (or opinion) text for low-resource language mixed with English (Chakravarthi et al., 2020a,b; Ravikiran and Annamalai, 2021; Hegde et al., 2022). While Nepali-English code-switching has been a growing phenomenon in Nepalese society, especially in social media, there is no sentiment analysis dataset focusing on code-mixed scenarios. Hence, for the study of the language preference for expressing sentiment in code-switched Nepali-English, we create a sentiment analysis dataset and perform the tests on our hypotheses.

## 3 Hypotheses

In this study, we attempt to address the research question: *"Do Nepali-English speakers have a preference for using native language while expressing Negative sentiment in social media?"* We inves-

tigate this phenomenon using the proportions of words from certain languages used to express certain sentiments.

We define two hypotheses to test in this study:

**Hypothesis I:** There is an association between sentiment and language proportions.

**Hypothesis II:** The proportion of Nepali language use is higher for negative sentences than positive sentences.

The first hypothesis attempts to test whether there is any relation between the proportions of language used for expressing sentiment in social media or not. If there is an association between those two, the next hypothesis will check if the proportion of Nepali language use is higher for negative sentences than for positive sentences. The second hypothesis attempts to test the pragmatic behavior of the Nepali-English multilingual speakers in social media.

## 4 Dataset

### 4.1 Data Collection

YouTube is one of the most popular social media platform. The number of videos targeted to Nepali audiences within the platform is also increasing. The comments on these videos mostly express the sentiments of the commentator(s). The study conducted by Ndubuisi-Obi et al. (2019) determine that the topics that relate to societal tensions (e.g., political and socio-economics) affect code switching strongly. Hence, for collecting the comments from YouTube, top 10 YouTube channels in Nepal under the category "News&Politics" were listed. All the comments and their threads from top 50 videos of each channel were extracted using YouTube API. No information regarding the commentators were collected. The comments with less than 4 tokens and the comments containing Devanagari scripts were filtered out. In order to filter the non code-mixed comments, the best performing language identification model from (Pahari and Shimada, 2023) with F1-score of 94.66 was used. This model predicts one tag for each token in the sentence out of five tags: English, Nepali, named-entity, others, and ambiguous. The English and Nepali token counts were used to calculate the Code Mixing Index (CMI) (Das and Gambäck, 2014) for each sentence using the Equation 1.

Table 1: Dataset statistics showing the number of comments in each split and their total.

|  | Positive | Neutral | Negative | Total |
|---|---|---|---|---|
| **Train** | 2,768 | 2,918 | 2875 | 8,561 |
| **Dev** | 346 | 365 | 360 | 1,071 |
| **Test** | 346 | 365 | 359 | 1,070 |
| **Total** | 3,460 | 3,648 | 3,594 | 10,702 |

$$CMI = \begin{cases} 100 * \left[1 - \frac{max(w_i)}{n-u}\right], & \text{if } n > u \\ 0, & \text{if } n = u \end{cases} \quad (1)$$

Where, $w_i$ is the number of words in language $i$, $n$ is the total number of tokens, and $u$ is the number of language independent tokens. The CMI measures the level of mixing between the languages in the corpus. In this study, this measure is utilized to obtain the level of mixing between the languages in a comment. The comments having CMI less than 20 are filtered out to ensure the mix of English and Nepali tokens in the dataset. Furthermore, the comments often contained personally identifiable information as person names. These names were anonymized by replacing random, yet real person names. The gender of names were maintained during the replacement.

### 4.2 Data Annotation

The pool of filtered comments was randomized for annotation. Similar to Patwa et al. (2020), annotators were asked to annotate each comment into three categories: Positive, Neutral, and Negative. Two annotators were initially assigned to annotate all the comments. Inter-rater reliability between the two annotators using Cohen's kappa (k) (Cohen, 1960) was calculated and found it to be 0.55, suggesting moderate agreement between the annotators. The third annotator reviewed the disagreements between the annotators and resolved them by consensus. Most of the disagreements were observed on the borderline cases between neutral and other two classes. For example, *"Background sound ali low garna paryo."* (English Translation: *"Background sound should be lowered")* was marked as negative by one, while neutral by the other. This review can be interpreted as a suggestion to lower the background volume and hence can fall into the category 'Neutral' while this can also be interpreted as 'Negative' emotion as the commentator was bothered by the background sound.

The annotators annotated 10,702 comments in total. The statistics of the annotated dataset are provided in Table 1. The dataset is publicly released to encourage the research on code-mixed sentiment analysis in Nepali-English language pair.

## 5 Baseline Classifiers

Traditional machine learning models and transformer-based models are applied for determining the sentiments from the Youtube comments as the simple baseline. The models used in this study are listed in this section.

### 5.1 Machine learning-based models

We consider classical machine learning techniques namely: Support vector machine (SVM) and multi-layer perceptron (MLP) with different embeddings. These models are implemented using the sklearn library (Pedregosa et al., 2011). A 'linear' kernel is used for SVM. The number of hidden layer size is set to two in case of MLP. The following embeddings are used with these classical techniques:

#### 5.1.1 TFIDF

Term frequency inverse document frequency (TFIDF) is a common algorithm to transform textual data into numerical representations. This method quantifies the significance of the words within the comments while considering their prevalence across the entire comments. This method is used in different NLP tasks due to its simplicity, interpretability, and computational efficiency.

#### 5.1.2 LASER

Language agnostic sentence representations (LASER) (Artetxe and Schwenk, 2019) is a contextualized language model that is based on BiLSTM encoder and is trained using multiple sources of publicly available parallel corpora using the translation objective. The LASER model was trained to generate the numerical representations for 93 languages, belonging to more than 30 different language families and written in 28 different scripts. Joint training in different languages makes this model leverage competitive performance in low-resource languages.

#### 5.1.3 LaBSE

Language agnostic BERT sentence embedding (LaBSE) (Feng et al., 2022) is a BERT-based cross-lingual sentence embedding model trained using

masked language modeling and translation language modeling objectives on translation ranking tasks. The LaBSE model supports 109 languages. LaBSE produces similar representations for the parallel sentences in different languages. This model has demonstrated strong performance even on languages in which the model was not trained exclusively.

### 5.2 Transformer-based Models

Apart from classical machine learning models, we conduct experiments with different transformer-based models as well. Transformer-based models are the current default methods in NLP field due to their high performance. The ability of multilingual transformer based models to produce aligned representations of multiple languages are beneficial for handling code-mixed text (Winata et al., 2021). The classification model consists of the pre-trained language model with a linear layer with dropout on top. The experiments are run using transformers library (Wolf et al., 2020). AdamW optimizer is used with the learning rate of $1e - 5$. The training is run for 5 epochs and best performing model in validation set is used for testing.

#### 5.2.1 mBERT

Multilingual BERT (mBERT) (Devlin et al., 2019) is the multilingual counterpart of BERT. mBERT is pre-trained on Wikipedia data from 104 languages. mBERT model is pre-trained with masked language modeling and next sentence prediction objectives. This model is able to produce cross-lingual representations which can be used for many multilingual tasks in NLP.

#### 5.2.2 XLM-R

XLM-RoBERTa (XLM-R) (Conneau et al., 2020) is a transformer model trained for masked language modeling using monolingual data in 100 languages with 2.5 TB of text. XLM-R model is the modified version of XLM (Lample and Conneau, 2019) that avoids translation language modeling and employs RoBERTa (Liu et al., 2019) instead of BERT. The performance of XLM-R is superior to mBERT on various cross-lingual benchmarks by 23% in accuracy in low-resource languages.

#### 5.2.3 MuRIL

Multilingual representation of Indian languages (MuRIL) (Khanuja et al., 2021) is an Indian subcontinent language family model which is pre-trained

on a large corpora of languages in Indian sub-continent. The model is pre-trained on 16 Indian subcontinent languages and English. Masked language modeling and translation language modeling objectives were used in the pre-training of this model. This model outperformed other multilingual models on the tasks involving Indian subcontinent languages. This model includes both Devanagari scripts and its transliterated form during the training.

## 6   Results and Discussion

### 6.1   Hypotheses Test

**Sentence Count:**   In order to identify the dominant language of each comment, we utilize the same language identification model as discussed in Section 4.1. For each comment, we identify the language tags for all tokens in the comment. We distinguish the dominant language of the comment utilizing the number of specific language tokens in the comment. If the number of English language tokens is greater than the number or Nepali language tokens in a comment, we consider the comment as an English comment and vice versa. When the number of English language tokens and Nepali language tokens are equal, we consider the comment as having no distinct language. The mosaic chart on Fig 1 shows the statistics for the number of sentences belonging to each sentiment class against the dominant language of the sentence. We use these statistics to run our statistical tests on the hypotheses explained in Section 3.

**Statistical Tests:**   In order to test the hypothesis I discussed in Section 3, we use the chi-squared test. This test is used to check the independence between two categorical variables. In our case, the variables are the dominant language and sentiment class. The null hypothesis for this test is: 'There is no association between sentiment and language', while our alternative hypothesis is Hypothesis I. Significance levels were set at 1% level. The p-value obtained from the test was significantly lower than our significance level. Hence, we reject the null hypothesis and accept the alternative hypothesis. In other words, this result supports our Hypothesis I, i.e., there is an association between sentiment and language.

Since there is an association between the sentiment classes and the dominant language of the comment, we test our second hypothesis to check

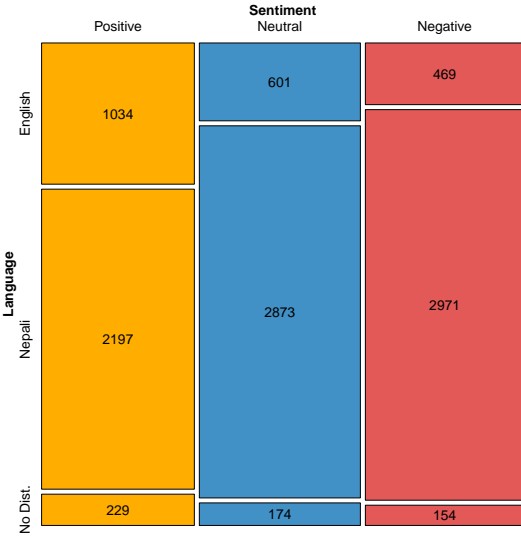

Figure 1: Mosaic chart showing the frequency of sentences in each language for each sentiment class for the human annotated comments.

if there is statistically significant difference in the proportion of use of Nepali language in different sentiment groups. We test the second hypothesis using the z-test for proportions. The z-test for proportions is used to test a hypothesis about the difference between the proportions of two samples. In our case, we take the proportions of Nepali language comments on positive class and on negative class. The null hypothesis for this test is: 'The proportion of Nepali language use is the same for negative sentiment and positive sentiment comments', while our alternative hypothesis is the Hypothesis II. After computing the z-test for proportions, we found that our z-value is significantly lower than -4, hence we can reject the null hypothesis and accept the alternative hypothesis. Hence, statistically we conclude that the proportion of Nepali language is higher in negative comments when compared with positive comments. Moreover, from the same test conducted for the proportions of English language in positive and negative comments we noticed the proportion of English language use was higher in positive than negative comments in our dataset.

These results reflect those of (Agarwal et al., 2017; Rudra et al., 2019) who also found that multilinguals prefer to express the emotions with their first language. Most of the multilingual people in Nepal learn their first language, Nepali at home. Whereas, their second language, English in schools (Gurung, 2019). Hence, most of the multilingual

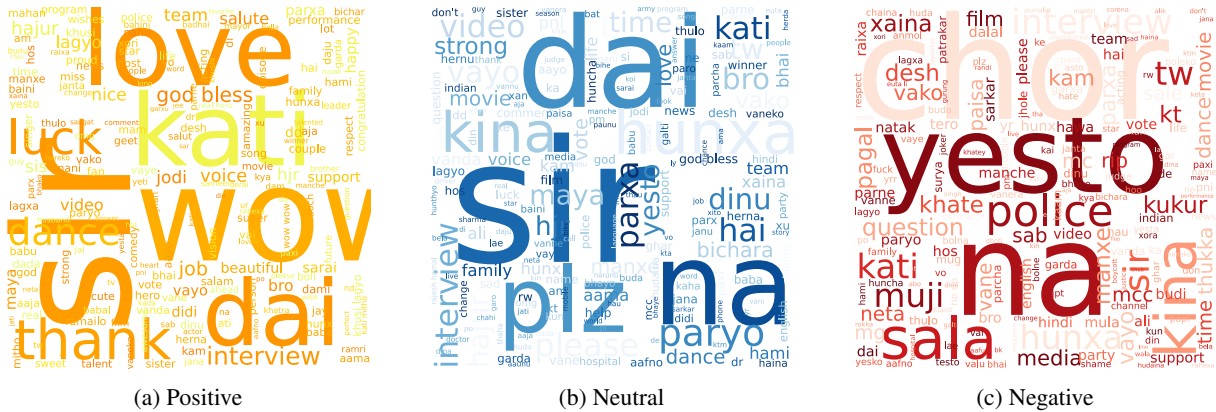

| (a) Positive | (b) Neutral | (c) Negative |

Figure 2: Wordclouds of comments across different sentiment classes.

speakers gain the knowledge of English as an instructed language. Therefore, this finding is consistent with that of Dewaele (2004) who discusses how instructed language learners have a limited general knowledge of negative words. As a result, the speakers tend to use the instructed language (i.e., English in context of this study) infrequently for expressing negative sentiments. Figure 2 presents the wordcloud of comments for each sentiment class in the dataset. It can be seen from the figures that more proportion English words can be seen on Positive wordcloud and more proportions of Nepali words can be seen on Negative wordcloud.

## 6.2 Sentiment analysis on CS

Table 2 and 3 shows the experimental results in terms of F1- score for the machine learning and transformer-based models respectively. The experiments are performed on the data split explained in Section 4.2. The average of three runs is reported on both the tables.

SVM model with TFIDF embedding produces the best result (0.68 for Macro-, Weighted-F1, and Accuracy) among the machine learning methods in the experiment. SVM model performs better than MLP for all the embeddings except LASER embedding where its performance is similar to that of MLP. LaBSE and LASER embeddings are the multilingual sentence embeddings that are trained to produce the semantically meaningful sentence representations by leveraging the neural networks and cross-lingual training. On the other hand, TFIDF is a simple model which computes the numerical representation based on the importance of the words within the document and across the entire corpus. Better performance by this simple method illustrates the added complexity for the models

trained in monolingual data in multiple languages, due to the mixing of the languages in our dataset. Our dataset consists of mixed Nepali-English data. Transliterated form of Nepali is used in the dataset which is not used during the training of the aforementioned models. Hence due to the mixing and the use of romanized script for Nepali language, the embeddings from the multilingual sentence embedders perform lower than TFIDF.

In case of transformer-based models, all three models perform in similar fashion. The highest performance is exhibited by MuRIL. All these models are trained on multiple languages together with the languages involved in our study: Nepali and English. While mBERT and XLM-R are trained on the monolingual data in these languages, MuRIL is trained on the monolingual data, parallel translated data, and the transliterated data. As discussed earlier, transliterated form is observed highly in informal settings like social media platforms and our dataset contains the transliterated form of Nepali language. Hence, MuRIL vocabulary takes into account higher percentage of tokens from our dataset as compared with mBERT and XLM-R, hence the performance is better for MuRIL. Few previous studies (Adhikari et al., 2022; Pahari and Shimada, 2023) demonstrated that the language family-specific models can provide significant benefit when fine-tuning training dataset size is of certain minimum number, which suggests that there is room for improvement for the performance by introducing more training dataset by some techniques like data augmentation.

Closer inspection of the table shows that both machine learning and transformer-based models demonstrated lower scores for neutral cases when compared against positive and negative cases. This

Table 2: Experimental results using machine learning-based models.

| Embedding | Model | Negative | Neutral | Positive | Macro F1 | Weighted F1 | Accuracy |
|---|---|---|---|---|---|---|---|
| TFIDF | SVM | 0.67 | **0.61** | **0.76** | **0.68** | **0.68** | **0.68** |
| | MLP | **0.68** | 0.56 | 0.74 | 0.66 | 0.66 | 0.66 |
| LaBSE | SVM | 0.62 | 0.55 | 0.72 | 0.63 | 0.63 | 0.63 |
| | MLP | 0.61 | 0.53 | 0.73 | 0.62 | 0.62 | 0.62 |
| Laser | SVM | 0.64 | 0.55 | 0.70 | 0.63 | 0.63 | 0.63 |
| | MLP | 0.62 | 0.54 | 0.73 | 0.63 | 0.63 | 0.63 |

Table 3: Experimental results using transformer-based models.

| Model | Negative | Neutral | Positive | Macro F1 | Weighted F1 | Accuracy |
|---|---|---|---|---|---|---|
| mBERT | 0.68 | 0.59 | 0.76 | 0.68 | 0.68 | 0.68 |
| XLM-R | 0.67 | 0.54 | 0.75 | 0.65 | 0.65 | 0.66 |
| MuRIL | **0.72** | **0.60** | **0.80** | **0.70** | **0.70** | **0.70** |

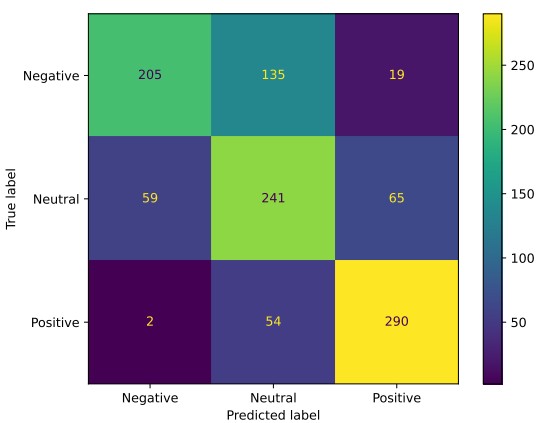

Figure 3: Confusion matrix of the result from the MuRIL model.

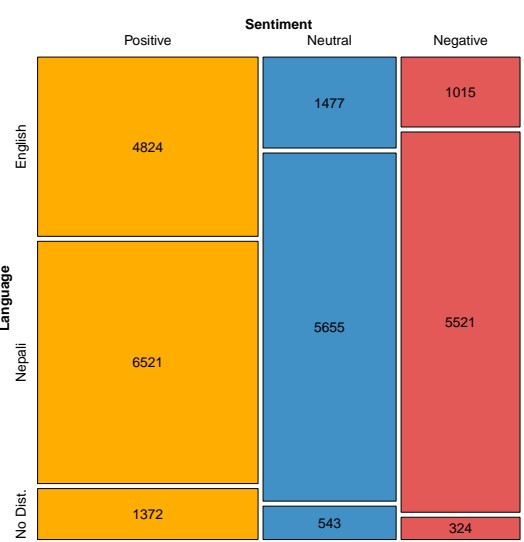

Figure 4: Mosaic chart showing the frequency of sentences in each language for each sentiment class for the larger pool of automatically annotated comments.

is due to borderline comments that are difficult even for humans as discussed in Section 4.2. This can be visualized in the confusion matrix for one run in the MuRIL model is shown in Fig. 3. 89.2% of wrong prediction of Negative classes were Neutral class and 83.1% of prediction of Positive class were Neutral class.

### 6.3 Hypothesis test on larger pool of automatically classified comments

Section 6.1 discussed the hypothesis test on the limited human annotated data. With the availability of automatic sentiment classifier as discussed in Section 6.2, further test is performed on the large pool of comments. The MuRIL-based classifier is utilized to automatically classify 27,252 unannotated comments collected in Section 4.1. The dominant language for each comment is determined using the same language identification model as

Section 6.1. The mosaic chart on Fig 4 shows the statistics for the number of comments belonging to each sentiment class against the dominant language of the comment. As visualized in the chart, the proportion of comments with Nepali as dominant language are higher for negative comments than for positive comments. Similar to the statistical tests on human annotated data, the statistical test performed on these automatically annotated data also validates both of our hypotheses.

### 6.4 Conclusion

In this study, we collected public comments and annotated them with sentiment annotations. With the

help of the newly created dataset, we test and accept two hypotheses. First hypothesis confirms the dependence between the language used in the comment and the sentiment of the comment. Second hypothesis confirms the higher proportions of Nepali comments observed in expressing negative sentiments as compared with positive sentiment. Similarly, the proportions of English is higher in positive sentiments than negative. The results aligns with the conclusion of previous studies (Agarwal et al., 2017; Rudra et al., 2019), preference of first language of the speakers for expressing sentiments or swearing. The results of machine learning methods show that the multilingual sentence embedders fail to generate proper representations for code-switched languages. Considerably more work will need to be done to generate multilingual embeddings that can capture the semantic meaning of mixed languages as well. Language identification model trained on code-switched data from Twitter was used in the analysis. However, the accuracy of the language identification model was not evaluated due to unavailability of test data for the Youtube domain. In future work, we need to evaluate the model accuracy on this domain, and verify the influence for our analysis. Furthermore, the findings raises few socio-lingustic questions about the influence of English language in Nepalese communities and its impact on Nepali language, which would also be a fruitful area for further work.

## Limitations

The dominant language for the comments is identified based on the number of language tokens, which are identified using automatic language identification model that might have non-negligible errors. We use these data as descriptive statistics and analyze the aforementioned hypotheses.

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
