# OpenReview forum: "Language Preference for Expression of Sentiment for Nepali-English Bilingual Speakers on Social Media"
_EMNLP/2023/Workshop/CALCS — EMNLP 2023 Workshop CALCS_

### Official Review · Reviewer_9n1Q · 2023-10-03
**The paper is well-written and employs experiments and analyses to address the research questions. Addressing concerns related to the representativeness of the dataset , language identification tools, and performing additional analyses on the larger dataset would further strengthen the paper.**

**Rating:** 4
**Confidence:** 4

**Review:**

The paper investigates an interesting and relevant question on the language preference of code-switching behaviour by bilingual Nepalese speakers in the context of expressing sentiment. Overall, the paper is well-written and presents experiments and analyses suitable for answering the research questions outlined by the authors. The paper will benefit from addressing the following comments:
- Potential bias in the dataset: How representative is the dataset used for this study? For example, how many distinct users wrote the comments in the dataset? What is the influence of topical focus of the Youtube channels from where the dataset was sampled on the reported findings?
- While the authors acknowledge the potential limitations of the language identification tools they used, a more practical step has to be taken to ensure that this limitation does not constitute a major threat to the validity of the reported findings.
- The authors contribute an annotated dataset sampled from a larger pool. All experiments and analyses were conducted on the annotated data. It would be interesting to also perform additional analyses on the larger pool using the model(s) trained on the annotated dataset. Perhaps, this may address my concern above on the threat to validity.

**copy editing comment**
line 209: "effect" -> "affect"

**Candidate For Best Paper:**

No

**Reason For Best Paper:**

Not a candidate for best paper.

**Related:**

5: It is very related to the workshop.

---

### Official Review · Reviewer_k8q6 · 2023-10-03
**The paper shows that bilingual speakers of nepali and english languages  use their first language  (nepali)  when expressing negative sentiments. For this they collected around 11K Youtube comments from top 10 Youtube channels in Nepal under News&Politics category. They also present baseline results for sentiment classification with traditional SVM, MLP and transformer based models.**

**Rating:** 3
**Confidence:** 5

**Review:**

The  authors explore the use of Nepali language to express negative sentiments in Youtube comments. In addition, they also provide baseline results on the Nepali-English code-switched dataset using traditional SVM, MLP, and more recent transformer based models.

Strength
- the Nepali-English code-switched sentiment dataset will be publicly available
- showed that bilingual people of Nepal mainly use Nepali language to express negative sentiment, which aligns with previous research
- baseline results on sentiment classification on code-switched Nepali-English sentiment dataset

Weakness
- the token level tagging was done through off-the-self model. Adding some analysis on how good or bad the system is on the YouTube comments datasets would be helpful. The model might have been trained on some other domain than Youtube.
- it would be great to add data analysis. (most frequently used nepali words/slangs in codeswitched context)
- model interpretation showing why TFIDF being one of the most simple method requiring less resources than transformer based models performed better.
- the paper mentions the effect of tokenization as potential source of error. it would be great to add detail analysis of impact of tokenization on sentiment classification
- it would be great to add LLM based experiments for sentiment classification

**Candidate For Best Paper:**

No

**Reason For Best Paper:**

N/A

**Related:**

5: It is very related to the workshop.